# The Junctophilin-2 Mutation p.(Thr161Lys) Is Associated with Hypertrophic Cardiomyopathy Using Patient-Specific iPS Cardiomyocytes and Demonstrates Prolonged Action Potential and Increased Arrhythmogenicity

**DOI:** 10.3390/biomedicines11061558

**Published:** 2023-05-27

**Authors:** Joona Valtonen, Chandra Prajapati, Reeja Maria Cherian, Sari Vanninen, Marisa Ojala, Krista Leivo, Tiina Heliö, Juha Koskenvuo, Katriina Aalto-Setälä

**Affiliations:** 1Heart Group, Faculty of Medicine and Health Technology, Tampere University, 33520 Tampere, Finland; 2Tampere University Heart Hospital, 33520 Tampere, Finland; 3Heart and Lung Center, Helsinki University Hospital, University of Helsinki, 00290 Helsinki, Finland; 4Blueprint Genetics, 02150 Espoo, Finland

**Keywords:** hypertrophic cardiomyopathy, junctophilin-2, disease modelling, isogenic human pluripotent stem cell-derived cardiomyocytes

## Abstract

Hypertrophic cardiomyopathy (HCM) is one of the most common genetic cardiac diseases; it is primarily caused by mutations in sarcomeric genes. However, HCM is also associated with mutations in non-sarcomeric proteins and a Finnish founder mutation for HCM in non-sarcomeric protein junctophilin-2 (JPH2) has been identified. This study aimed at assessing the issue of modelling the rare Finnish founder mutation in cardiomyocytes (CMs) differentiated from iPSCs; therefore, presenting the same cardiac abnormalities observed in the patients. To explore the abnormal functions in JPH2-HCM, skin fibroblasts from a Finnish patient with JPH2 p.(Thr161Lys) were reprogrammed into iPSCs and further differentiated into CMs. As a control line, an isogenic counterpart was generated using the CRISPR/Cas9 genome editing method. Finally, iPSC-CMs were evaluated for the morphological and functional characteristics associated with JPH2 mutation. JPH2-hiPSC-CMs displayed key HCM hallmarks (cellular hypertrophy, multi-nucleation, sarcomeric disarray). Moreover, JPH2-hiPSC-CMs exhibit a higher degree of arrhythmia and longer action potential duration associated with slower inactivation of calcium channels. Functional evaluation supported clinical observations, with differences in beating characteristics when compared with isogenic-hiPSC-CMs. Thus, the iPSC-derived, disease-specific cardiomyocytes could serve as a translationally relevant platform to study genetic cardiac diseases.

## 1. Introduction

HCM is a cardiac condition typically caused by sarcomere gene variations leading to thickened intraventricular septum among other things. Cardiomyocytes are hypertrophied, disorganized, and separated by areas of interstitial fibrosis at the cellular level. HCM is geno- and phenotypically heterogenous disorder is subject to significant morbidity and mortality in patients of all ages, with an estimated clinical prevalence of 1:500 in the general adult population [1]. However, in 5–10% of adults and 25% of children diagnosed with HCM, a non-sarcomeric form of disorder is found [2]. Non-sarcomeric forms of HCM include various metabolic, neurodegenerative, and storage diseases [3], such as Fabry’s disease [4], Danon’s disease [5], and Friedreich’s ataxia [6]. Several Finnish patients diagnosed with HCM were found to carry a non-sarcomere gene JPH2 mutation in a study by Vanninen [7]. The carriers of the mutations have significant septal thickening and a high prevalence of severe arrhythmias that results in most of the mutation carriers having an intracardiac defibrillator (ICD).

In mature cardiac and skeletal muscle cells, efficient excitation–contraction coupling (ECC) and integrity of the junctional membrane complex (JMC) are arranged by junctophilin-2 (JPH2) in the proximity of transversal tubules (T-tubules) [8,9,10] that are still poorly developed in neonatal cardiomyocytes [11,12]. In fact, JPH2 enables the proper plasma membrane (PM) and endo-/sarcoplasmic reticulum (ER/SR) coupling in cardiac and skeletal muscle cells and a common finding in failing hearts is the disruption of the JMC and downregulation of JPH2 [8,10,13]. Within the JMC, striated muscle preferentially expresses protein kinase (SPEG) that directly binds to JPH2 and ryanodine receptor type 2 (RyR2) [14], which also has a critical function in regulating the ECC (Figure 1a). Junctophilin protein binds to L-type Ca^2+^ channels on the cell membrane and ryanodine receptors in the sarcoplasmic reticulum (SR) and thus enables the proximity of these two Ca^2+^ channels and a controlled way to trigger Ca^2+^ release from the SR for optimal function of CMs (calcium-induced calcium release, CICR). Mutations in JPH2 have been associated with HCM; this was originally found in 2007 in four probands among Japanese HCM patients with p.(Gly505Ser) [15]. Since then, multiple studies have linked JPH2 mutations to HCM [8,16,17,18,19,20,21].

Modeling rare diseases benefits from the ability of patient-specific hiPSCs to propagate indefinitely while having the potential to differentiate into any human cell type. This has enabled studies involving tissues such as neurons and cardiomyocytes that have been previously unavailable [22]. For example, Fabry’s disease, which results from enzyme deficiency, has seen new insights in disease mechanisms and drug interactions after studies with varying mutations causing the disease [23,24]. Robust hiPSC phenotypes in disease models that reflect patient symptoms can also bring new indications for drugs that have already been approved for other diseases and gene therapies for the potential treatment of patients. Validation of hiPSC results using isogenic controls generated with genome editing is essential for rare diseases, especially where clinical efficacy of treatment is not predictive in current animal models [25].

Current research has also widely applied hiPSC-CMs in the study of HCM pathogenesis and characterization of the phenotype using standard protocols in cellular morphology, electrophysiology, calcium handling, and metabolism. In addition, several groups have introduced or repaired HCM-related mutations using gene editing technologies [26]. Here, we derived hiPSCs from an HCM patient carrying a Finnish founder mutation in the *JPH2* gene. We reprogrammed the patient-derived fibroblasts to hiPSCs, corrected the mutation using a clustered regularly interspaced short palindromic repeats (CRISPR/Cas9) genome editing method to create isogenic hiPSCs, differentiated the two patient-specific hiPSCs into CMs, and compared the phenotypes of the T161K and isogenic hiPSC-CMs.

This research addresses the issue of modelling the rare Finnish founder mutation in CMs differentiated from hiPSC and therefore presenting the same cardiac abnormalities observed in patients. Equally, we created an isogenic iPS cell line with a corrected cardiomyocyte cellular phenotype. Comparison of mutated (T161K) and mutation-corrected (isogenic) cardiac cell lines provides an understanding of whether the correction of disease-causing mutations can be used in in vitro cardiotoxicity studies or cell-based therapeutics in the future. This model can be used in further studies to better reflect the possible effects of current HCM treatments in humans.

## 2. Materials and Methods

### 2.1. Clinical Data of the Patient

We derived iPSCs from a patient heterozygous for a HCM causing variant in JPH2-Thr161Lys (Figure 1b). The male patient was diagnosed with HCM at the age of fourteen with thickening of the septum (16 mm); β-blocker treatment was prescribed. At the age of sixteen, further thickening of the septum (28 mm) and increased LVOT gradient was observed. These findings were confirmed with cardiac MRI. β-Blocker treatment was continued with the addition of disopyramide. Due to frequent non-sustained ventricular tachycardia (NSVT) episodes with maximal β-blocker treatment, an ICD was implanted at the age of 23. Due to dyspnea and increased LVOT gradient at stress, alcohol ablation was performed at the age of 24.

### 2.2. Generation and Culture of the Patient-Specific hiPSC Line

The hiPSC cell line 09703.HCMJp was generated from skin fibroblasts with the plasmid vectors OCT4, KLF4, c-MYC, and SOX2 using nucleofection (Lonza #V4XP-2024) according to the manufacturer’s instructions. The cell line was derived and cultured on mouse embryonic fibroblast (MEF) feeder cell layers (26,000 cells/cm^2^, CellSystems Biotechnologie Vertrieb GmbH, Troisdorf, Germany) in a human pluripotent stem cell (hPSC) culture medium consisting of KnockOut DMEM (KO-DMEM, Gibco, Life Technologies Ltd., New York, NY, USA) supplemented with 20% KnockOut serum replacement (ko-SR, Gibco, Life Technologies Ltd., New York, NY, USA), 1% nonessential amino acids (NEAA, Lonza Group Ltd., Basel, Switzerland), 2 mM GlutaMAX (Gibco, Life Technologies Ltd., New York, NY, USA), 50 U/mL penicillin/streptomycin (Lonza Group Ltd.), 0.1 mM 2-mercaptoethanol (Gibco, Life Technologies Ltd., New York, NY, USA), and 4 ng/mL basic fibroblast growth factor (bFGF, PeproTech, Rocky Hill, NJ, USA).

### 2.3. CRISPR/Cas9 Genome Editing to Generate Isogenic Control Cell Line

To validate the molecular and cellular phenotype of the JPH2-Thr161Lys mutation, we targeted the JPH2 gene locus of the hiPSC cell line 09703.HCMJp using CRISPR-Cas9 genome editing technology. The CRISPR knock-in strategy for the generation of the isogenic line was designed on a plasmid-based system employing pLCas9 (Addgene #44719). Potential Cas9 target sites close to the mutation site were identified using an online CRISPR design tool (crispr.mit.edu; 2 April 2020) [27]. Four sgRNAs were selected and cloned into a U6-driven gRNA expression vector (Addgene #41824) as previously described [28,29].

The level of activity of each sgRNA was validated using the T7EI assay in HEK cells, and the best guide RNA (TACGCCAGAGCGTGCCCTAC) was selected for the 09703.HCMJp isogenic line generation. As the repair template, a dsDNA HDR (homology-directed repair) fragment of 3 kbp containing the corrected base in the middle of sequence was amplified from the genomic DNA of a healthy hiPSC cell line. The amplified HDR fragment was cloned into the pJET1.2/blunt using the CloneJET PCR cloning kit (Thermo Fisher Scientific, Waltham, MA, USA).

Gene editing was performed as previously described, with slight modifications. Briefly, the human iPS cells that were expanded on Geltrex in feeder-conditioned medium were pre-treated with 10 mM ROCK inhibitor (Tocris Bioscience, Bristol, UK) in mTeSR1 before nucleofection. Cells were digested with Versene to form a single-cell suspension, followed by centrifugation for 3 min at 110× *g*. After harvesting, 2 × 10^6^ cells were nucleofected (Amaxa 4D-Nucleofector X Unit, program CB-150) using a P3 Primary Cell 4D-Nucleofection Kit (V4XP-3032, Lonza, Basel, Switzerland). Each nucleofection reaction included 2 μg of guide plasmid, 2 μg of donor plasmid, and 2 μg of Cas9 expressing plasmid. The nucleofected cells were reseeded into a Geltrex-coated 100 mm culture plate, and 48 h after nucleofection the cells were treated with G418 antibiotic (100 μg/mL) for 24 h. After treatment, the cells were given fresh culture medium. Individual colonies were picked after two weeks and were expanded to validate the corrected isogenic lines by sequencing. The pluripotency of the corrected isogenic line was verified by PCR and immunocytochemistry as described below.

### 2.4. Characterization of iPS Cells

#### 2.4.1. Mutation Analysis

The TaqMan sample-to-SNP Kit (Applied Biosystems, Life Technologies Ltd., Waltham, MA, USA) was used to prepare DNA samples from the hiPSC cell line, and the presence of Thr161Lys mutation was confirmed by custom TaqMan SNP genotyping assays (Applied Biosystems, Life Technologies Ltd., Waltham, MA, USA) according to manufacturer’s instructions. Specific primers were used to amplify the JPH2 gene in the genotyping assays.

#### 2.4.2. Expression of Mutant and Wild-Type Alleles in hiPSC-Derived CMs

RNA samples were collected and extracted from hiPSC-derived CMs (09703.HCMJp) with Norgen’s Total RNA Purification Plus Kit (Norgen Biotek Corp., Thorold, ON, Canada) according to the manufacturer’s instructions. A total of 50–100 ng of RNA was transcribed to cDNA using the High-Capacity cDNA Reverse Transcription Kit (Applied Biosystems, Life Technologies Ltd., Waltham, MA, USA). The expression of the Thr161Lys mutation at the mRNA level in the hiPSC-derived CMs was studied using custom TaqMan SNP genotyping assays (Applied Biosystems, Life Technologies Ltd., Waltham, MA, USA) as above.

#### 2.4.3. Immunocytochemistry

Undifferentiated hiPSC colonies were fixed with 4% paraformaldehyde (PFA, Sigma-Aldrich, St. Louis, MO, USA), stained with primary antibodies for Nanog (R&D Systems Inc., Minneapolis, MN, USA), OCT4 (R&D systems Inc., Minneapolis, MN, USA), SOX2 (Santa Cruz Biotechnology, Santa Cruz, CA, USA), TRA-1-60 (Millipore, Burlington, MA, USA), and TRA-1-81 (Millipore, Burlington, MA, USA), and visualized with secondary antibodies. Finally, the cells were mounted with Vectashield (Vector Laboratories Inc., Burlingame, CA, USA) containing 40,6-diamidino-2-phenylindole (DAPI) for the nuclear staining and imaged with an Olympus IX51 phase-contrast microscope equipped with fluorescence optics and an Olympus DP30BW camera (Olympus Corporation, Hamburg, Germany).

#### 2.4.4. RT-PCR

The RNA was extracted from the hiPSC lines using the NucleoSpin RNA II Kit (Macherey-Nagel GmbH & Co., Duren, Germany), and 500–1000 ng of RNA was transcribed to cDNA with the High-Capacity cDNA Reverse Transcription Kit (Applied Biosystems, Life Technologies Ltd., Waltham, MA, USA). The presence of the pluripotency genes NANOG, SOX2, REX1, OCT4, and MYC and the absence of virally imported exogenes (OCT4, SOX2, MYC, and KLF4) were confirmed by RT-PCR. GAPDH was used as an endogenous control. The primer sequences for pluripotency genes and virally imported exogenes have been published earlier [30]. The primer sequences used for detection of Sendai transgenes are described in the CytoTune-iPS Reprogramming Kit manual (Life Technologies Ltd., New York, NY, USA).

#### 2.4.5. Karyotype and Pluripotency Analysis

The karyotypes of hiPSC lines were studied by KaryoLite assay [31] (Turku Centre for Biotechnology, University of Turku, Turku, Finland). The pluripotency of hiPSC lines was confirmed in vitro by embryoid body (EB) formation. hiPSCs were removed from the feeder cell layer and cultured in suspension to form EBs. The EBs were cultured in KSR medium ((KnockOut DMEM containing 10% KnockOut Serum Replacement (Gibco, New York, NY, USA)), 1% MEM NEAA (Gibco, New York, NY, USA), 1% GlutaMAX (Gibco, New York, NY, USA), 0.2% β-mercaptoethanol (Gibco, New York, NY, USA), and 0.5% penicillin/streptomycin (Lonza, Basel, Switzerland)) for 4–6 weeks before RNA extraction. A total of 200 ng of RNA was transcribed to cDNA for the RT-PCR analysis. The presence of all three germ layers, endoderm (AFP, b-actin), mesoderm (alpha-cardiac actin, KDR), and ectoderm (SOX1, PAX6), was studied with RT-PCR.

#### 2.4.6. Differentiation of Cardiomyocytes

The hiPSC cell line 09703.HCMJp was differentiated into CMs by coculturing with mouse visceral endodermal-like cells (END2, Hubrecht Institute, Utrecht, The Netherlands). After 30–45 days, beating areas were cut from cocultures and dissociated into single cells in EB medium using Collagenase A (Roche Diagnostics, Mannheim, Germany) and plated to 0.1% gelatin-covered coverslips for further analysis.

### 2.5. Characterization of hiPSC-Derived Cardiomyocytes

#### 2.5.1. Immunocytochemistry

For all fluorescence microscopy analyses, 5 × 10^4^ cells were plated on a 12 mm-diameter glass coverslip. CMs were fixed with 4% PFA and stained with cardiac Troponin T (cTnT, 1:2000, ab64623, Abcam, Cambridge, MA, USA) and JPH2 (Junctophilin-2, 1:200, sc-377086, Santa Cruz Biotechnology, Dallas, TX, USA) primary antibodies, followed by labeling with secondary antibodies. Staining of nuclei was achieved with Vectashield antifade mounting medium with DAPI (Vector Laboratories, Burlingame, CA, USA). Images were obtained with a Zeiss Axio Scope.A1 upright fluorescence microscope with a Plan-Apochromat 20× objective with a numerical aperture of 0.8 and Zen 2010 software (Carl Zeiss, Frankfurt am Main, Germany). In certain images, the brightness was changed linearly with an open source image processing package Fiji (1.52n), that is based on ImageJ (NIH, Bethesda, MD, USA) [32].

#### 2.5.2. Analysis of hiPSC-CM Size and Nuclear Area

The size of cTnT and DAPI-stained CMs and their nuclear areas was analyzed from CMs in each cell line by application of the Otsu thresholding method [33] to the acquired fluorescence images and by measurement of area fractions in pixels with Fiji 1.52n software. Thresholds and brightness adjustments were always applied equally to images captured from T161K and isogenic hiPSC-CMs.

#### 2.5.3. Sarcomere Orientation

The orientation of the cell actin network in T161K and isogenic hiPSC-CMs was analyzed from fluorescence images with the CytoSpectre 1.2 spectral analysis tool [34]. In short, default settings were used for image analysis; however, image magnification (40×) and the pixel size of the camera (6.45 µm) were specified. The mixed component from the green channel, representing Alexa Fluor 488 secondary antibody against cTnT, was analyzed from the images. Cells with circular variance exceeding 0.9 were excluded from the rest of the analysis. The software was used to calculate the circular variance of the cellular orientation, describing the isotropy of orientation distribution in the image. Circular variance is bounded in the interval [0, 1], where a value closer to zero signifies distribution along the same direction (anisotropy) and a value closer to one designates spread distribution (isotropy).

#### 2.5.4. Electrophysiology

Action potentials (APs) from spontaneously beating hiPSC-CMs were recorded using a perforated patch with the help of Amphotericin B (Sigma-Aldrich, St. Louis, MO, USA) at a final concentration of 0.24 mg/mL, as previously described [35]. Current-clamp recordings were digitally sampled at 20 kHz and filtered at 2 kHz using a low pass Bessel filter on the recording amplifier (all from Molecular devices, San Jose, CA, USA). In addition, hiPSC-CMs were maintained at −70 mV by applying external current through a patch pipette, and APs were stimulated at 1 Hz using 2–4 ms depolarizing current pulses. The extracellular solution consisted of (in mM) 143 NaCl, 4.8 KCl, 10 HEPES, 5 D-glucose, 1.8 CaCl_2_, and 1.2 MgCl_2_ (pH adjusted to 7.4 with NaOH) (Sigma-Aldrich, St. Louis, MO, USA). The extracellular solution was preheated to 36 ± 1 °C. The patch electrodes (Harvard Apparatus, Cambridge, MA, USA) had a tip resistance of 2.5–3.5 MΩ, with the intracellular solution containing (in mM): 132 KMeSO_4_ (MP Biomedicals, Lowell, CA, USA), 4 EGTA (Sigma-Aldrich, St. Louis, MO, USA), 20 KCl, 1 MgCl_2,_ and 1 CaCl_2_ (pH adjusted to 7.2 with KOH (Sigma-Aldrich, St. Louis, MO, USA)). The recorded spontaneous APs and stimulated APs were analyzed using custom-made software in OriginLab (OriginLab 2018b, Amherst, MA, USA) and Clampfit 11 software (Molecular Devices, San Jose, CA, USA), respectively.

After recording the Aps, ionic currents were recorded from the same cells in a voltage-clamp technique using the same extracellular and intracellular solutions as described above. The Ca^2+^ currents (ICas) were recorded using a holding potential of −40 mV. The ICas were recorded in response to 300 ms test potentials from −50 mV to 50 mV with step size of 5 mV. The cell capacitances (Cms) were calculated by dividing the integral area of capacitive transient in response to 5 mV hyperpolarizing pulse from −80 mV. The peak ICas were divided by the Cm and presented as current density (pA/pF) to compensate for variations in cell size.

For the voltage dependence inactivation of a Ica, currents were determined at 10 mV (400 ms), preceded by a conditional step of 3 s ranging from −50 mV to 10 mV. Steady-state inactivation curves were fitted by using a Boltzmann equation: I/Imax = A/{1.0 + exp[(V1/2 − V)/k]}, in which V1/2 was half-maximum inactivation potential and k was the slope factor. The time course of recovery from inactivation was studied using the double-pulse protocol. For this, 300 ms of conditioning pulse (RP1) was applied to 10 mV. Then, the test pulse (P2) of 300 ms to 10 mV was applied after a recovery time (5, 10, 20, 50, 100, 200, 300, 400, 500, 600, 700, 800, 900, 1000, and 1500 ms) that varied between each recovery potential. The peak ICas elicited by RP2 (RI2) were normalized with peak ICas elicited by RP1 (RI1) and plotted as a function of the recovery interval. The curves were fitted using biexponential functions and fast (tf) and slow (ts) time constants were calculated.

The ICa was also recorded using the single voltage-clamp protocol with a holding potential of −80 mV, a prepulse to −40 mV for 50 ms, and then a test potential to 10 mV of 300 ms long. To study the ICa inactivation time course, the decreasing ICa of 90 ms during the test potential were fitted using monoexponential function and time constant (tau) were calculated. The peak ICa were divided by Cm and presented as current density (pA/pF).

Moreover, the action potential clamp (APC) technique was also used to study the activation, reactivation, and inactivation time constant of ICa by using a two-pulses protocol. For this, previously recorded AP was modified with a 50 ms prepulse to −40 mV from an HP of −80 mV. The depolarizing pulse (P1) was derived from the upstroke phase of AP, and the second pulse (P2) was initiated either from −40 mV (protocol 1) and −60 mV (protocol 2), respectively, to +10 mV. The activated ICas elicited from P1 (I1) and reactivated ICas elicited from P2 (I2) were normalized with cell capacitances and presented as current densities (pA/pF). Furthermore, the reactivation (I2) was also normalized to the previously activated ICas (I2/I1). The decreasing ICas elicited by P1 of 90 ms were fitted using a monoexponential function and the time constant (tau) of inactivation was calculated. All the recorded currents were analyzed by using Clampfit 11 software (Molecular Devices). All the electrophysiological experiments were performed at 36 ± 1 °C.

#### 2.5.5. Video Microscopy

Video microscopy was used to record videos of spontaneously beating single T161K and isogenic hiPSC-CMs. Monochrome videos were recorded for 30–60 s (720 × 480 resolution, 30 fps) after dissociation using video microscopy (Nikon Eclipse TS100, Nikon Corporation, Tokyo, Japan) with a video camera Optika Digi-12 (Optika Microscopes, Ponteranica, Italy). Normal beating was determined if the rhythm was regular and the contraction and relaxation phases followed each other without any delay or additional movement of the cell.

#### 2.5.6. Beating Analysis

Analysis of T161K and isogenic hiPSC-CM beating was performed using a digital image-based correlation (DIC)-based analysis method that was custom made in MatlabR2019a (MathWorks, Inc., Natick, MA, USA) software as described before [36]. Out of the signals obtained from a CM, the average parameters of beating were defined as: (1) duration of contraction, (2) time when contracted, and (3) duration of relaxation.

### 2.6. Statistical Analysis

For statistical analysis, IBM SPSS Statistics for Windows (Version 26.0, IBM Corp., New York, NY, USA) and GraphPad Prism Version 5.02 were used. An independent sample Mann–Whitney U test was used to analyze the statistical significance between the mutant and isogenic groups for data extracted from immunostaining and video analysis. Statistical comparisons of patch clamp data were made using unpaired or paired *t*-tests for unpaired and paired samples, respectively. The Pearson correlation coefficient (r) was calculated to measure the strength and direction of the relationship between two variables. *p* < 0.05 were considered as statistically significant. The data are presented as mean SEM for n cells.

## 3. Results

### 3.1. Characterization of iPS Cells

The pluripotent characteristics of the cell line were assessed, and results are presented in Figure 2. All the lines formed colonies that expressed proteins and genes typical for hiPSCs. The virally transferred endogenous genes were turned on and exogenous genes were silenced. The karyotypes of the hiPSC lines were normal.

### 3.2. Generation of Isogenic Cells by CRISPR/Cas9 Genome Editing

To understand the cellular consequences of the JPH2-T161K mutation and to compare its true impact in HCM, we utilized the isogenic hiPSC-line that had an identical genetic background and differed only in a single genetic change from the patient line. The results are presented in Figure 3. CRISPR/Cas9 was used to correct the JPH2 p.(Thr161Lys) mutation in the patient cell line, which was confirmed by sequencing. This isogenic cell line validated the hypertrophic phenotype, as mutant hiPSC-CMs displayed the main hallmarks of HCM such as an increase in cell size, sarcomeric disarray, and prolonged action potential.

### 3.3. Differentiation of Patient-Specific iPSCs into Cardiomyocytes

The END2 differentiation protocol was used (as described in [37,38]) to differentiate iPSCs into cardiomyocyte lineages (hiPSC-CMs). Both T161K and isogenic hiPSC-CMs maintained spontaneous contraction after dissociation into single hiPSC-CMs.

### 3.4. Morphology and Sarcomere Orientation of hiPSC-CMs

Cellular enlargement, sarcomeric disarray, and multinucleation are cellular features present in hypertrophic hiPSC-CMs six weeks after the induction of cardiac differentiation [39,40]. After differentiation into cardiomyocytes, beating clusters were dissociated into single cells and cultured for 6 weeks to determine whether the JPH2-Thr161Lys mutation initiates cardiac hypertrophic remodeling. Furthermore, we calculated the difference in cell and nucleus size, multinucleation, and orientation of sarcomeres between cTnT-positive T161K and isogenic hiPSC-CMs. Figure 4a visually represents the size difference between T161K and isogenic hiPSC-CMs, and qualitative analysis revealed a significant a 2.6-fold increase in cellular size (Figure 4b) (*** *p* < 0.0005, Mann–Whitney U test) and a 1.3-fold increase in individual nucleus size (Figure 4c) (*** *p* < 0.0005, Mann–Whitney U test). In addition, T161K hiPSC-CMs had an increased fraction of multinucleated CMs (Figure 4d,e). In addition, T161K hiPSC-CMs had significantly increased circular variance (** *p* < 0.002, Mann–Whitney U test, Figure 4f,g).

### 3.5. Action Potential and Calcium Current Properties

The action potentials (APs) were recorded in gap-free mode from the spontaneously beating hiPSC-CMs from both T161K and isogenic hiPSC-CMs. AP duration at 50% repolarization (APD50) and 90% repolarization (APD90) and maximum upstroke velocity (dVdt) were subsequently calculated. The hiPSC-CMs were categorized as ventricular-like hiPSC-CMs when APD90/APD50 < 1.35. Only ventricular-like hiPSC-CMs were compared between T161K and isogenic hiPSC-CMs. Figure 5a shows the representative AP traces from T161K and isogenic hiPSC-CMs. Our results showed that the average APD50 and APD90 values were significantly longer in T161K hiPSC-CMs (* *p* < 0.05, unpaired *t*-test) (Figure 5b,c). The average dVdt was similar in both T161K and isogenic hiPSC-CMs (Figure 5d). Both T161K and isogenic hiPSC-CMs were also stimulated at 1 Hz (Figure 5e); the results showe that the APD values at −20 mV and −60 mV were significantly longer in T161K hiPSC-CMs (Figure 5f,g). However, dVdt was similar between groups (Figure 5h). Figure 5i,j show the representative trace of AP exhibiting phase 3 EAD and phase-3-EAD-induced triggered arrhythmias. The occurrence of phase 3 EADs during the spontaneous beating was only observed in T161K hiPSC-CMs (Figure 5k) and was higher during the stimulation at 1 Hz (Figure 5l). 

The calcium currents (ICas) were recorded, and peak currents were normalized with cell capacitances to compensate for the variations in cell size and presented as current densities (pA/pF). Figure 6a shows the representative current densities–voltage plot; mean current densities displayed no significant changes at any test potentials. The voltage-dependent inactivation of ICas was also calculated and neither mid-potential (X = 0) nor slope (dx) showed any significant difference between T161K and isogenic hiPSC-CMs (Figure 6b). Furthermore, the time dependance of recovery from ICa inactivation was also studied using the double-pulse protocol, and it was found that both fast (τf) and slow (τs) time constants were similar in T161K and isogenic hiPSC-CMs (Figure 6c). The ICa current densities were also studied using the single voltage-clamp protocol with the holding potential at −80 mV (Figure 6d). The cell sizes were also compared and confirmed the previous finding that the cell size of T161K hiPSC-CMs was significantly larger than isogenic hiPSC-CMs (* *p* < 0.05, unpaired *t*-test; Figure 6e). The ICa current densities at 10 mV obtained from the single voltage protocol were similar between T161K and isogenic hiPSC-CMs. However, the inactivation kinetics of the ICa was slower in T161K hiPSC-CMs (Figure 6g). Since AP and ICa were recorded from same cells, the correlation test between APDs and ICa current densities as well as APDs and time constants of ICa inactivation were performed. Our results showed that no correlation between APDs and ICa current densities, which suggests that APDs were not dependent on ICa current densities (Figure 6h,i). Furthermore, we found a positive correlation between APDs and the time constants of ICa inactivation, suggesting the possible mechanism behind the longer APD in T161K hiPSC-CMs was because of slower inactivation of ICas (Figure 6j,k). 

We also incorporated the action potential clamp (APC) technique to study ICa activation during the upstroke phase of an AP and ICa reactivation with respect to previous ICa activation. Firstly, a previously recorded single AP was modified where the second depolarizing step pulse originated at −40 mV, as shown in Figure 7a. Figure 7b shows the first APC protocol and corresponding ICa traces from T161K and isogenic hiPSC-CMs. Secondly, the same AP was modified where the second depolarizing step pulse originated at −60 mV, as shown in Figure 7c. Figure 7d shows the second APC protocol and corresponding ICa traces from T161K and isogenic hiPSC-CMs. Our results show that current densities (I1 and I2) and I2/I1 in response to P1 and P2 from the first and second APC protocols are not different between T161K and isogenic hiPSC-CMs (Figure 7(e1−e3,f1−f3)). However, the time constant from the monoexponential fitting of the ICa in response to P1 from both APCs were significantly higher in T161K hiPSC-CMs (* *p* < 0.05, unpaired *t*-test; Figure 7(e4,f4). We also examined the correlation of I1 from the first APC protocol with APD and the ICa inactivation time constant since these parameters were recorded from the same hiPSC-CMs. Our results show no correlation between I1 and APD (−20 mV); however, there was a positive correlation between the ICa inactivation time constant and APD (−20 mV) in both T161K hiPSC-CMs and isogenic hiPSC-CMs. These results further support the fact that APD did not depend on ICa current densities but depended on ICa inactivation, i.e., the slower the ICa inactivation, the longer the APD.

We investigated further to better understand the mechanism behind the phase 3 EAD in T161K hiPSC-CMs. Figure 8a,b show the representative traces of the presence and absence of the AP exhibiting phase 3 EAD, respectively. First, we subdivided the APD (−20 mV) and APD (−60 mV) of AP from T161K hiPSC-CMs with and without the presence of phase 3 EAD, namely T161K^EAD+^ and T161K^EAD+^, respectively. Our results showed APD (−20 mV) and APD (−60 mV) were not significantly different between these two groups, which suggests that the occurrence of phase 3 EAD did not depend on APD (Figure 8c,d). Furthermore, ICa current densities and their inactivation time constants obtained from APC were also subdivided into two groups. The ICa current densities and their ratios were not significantly different between groups (Figure 8e–g, from APC protocol 1); however, the ICa inactivation time constants of T161K hiPSC-CMs exhibiting phase 3 EAD were significantly longer than those not exhibiting phase 3 EAD (** *p* < 0.01, unpaired *t*-test, Figure 8h, from APC protocol 1). Similar results were also found when comparisons were performed from APC protocol 2 (data not shown). These results suggest that slower ICa inactivation manifests the occurrence of phase 3 EAD in T161K hiPSC-CMs.

### 3.6. Beating Properties of hiPSC-CMs

We visually analyzed the beating behavior of single hiPSC-CMs in videos obtained from T161K and isogenic hiPSC-CMs by calculating contraction duration (Phase 1), duration contracted (Phase 2), relaxation duration (Phase 3), total contraction time (Phases 1–3), and BPM. The displacement model illustrated in Figure 9a was used to analyze the video recordings in detail. Figure 9b shows the representative beating amplitudes from T161K and isogenic hiPSC-CMs. Our results presented in Figure 9c normal beating in phase 1 (contraction) and phase 2 (contracted) with both cell types. However, in T161K hiPSC-CMs a significantly prolonged relaxation phase was observed (*** *p* < 0.001, Mann–Whitney U test). No significant differences in BPM were found between T161K and isogenic hiPSC-CMs (Figure 9d).

We also plotted the absolute time of contraction (phase 1) and relaxation (phase 3) against the beating rate (Figure 9e,f). We observed that a range of 150–400 ms corresponded to the contraction time of the isogenic hiPSC-CM cell population (Figure 9e). Generally, the contraction time was not affected by the beating rate; however, we observed a significant decrease in the contraction time in T161K hiPSC-CMs (* *p* < 0.05, Mann–Whitney U test) when the BPM increased (Figure 9e). Similar to the human heart [41], the beating rate decreased when the relaxation time increased, but the T161K hiPSC-CMs presented significantly longer relaxation times at all BPMs when compared with isogenic hiPSC-CMs (Figure 9f).

## 4. Discussion

In the present study, we generated hiPSCs from an HCM patient carrying a missense mutation p.(Thr161Lys) in the JPH2 gene. The JPH2 p.(Thr161Lys) mutation was further corrected by CRISPR/Cas9 to generate an isogenic hiPSC line. We found that T161K hiPSC-CMs displayed key HCM cellular hallmarks with increased cell and nucleus sizes accompanied by sarcomeric disarray. Additionally, T161K hiPSC-CMs displayed prolonged AP durations, a higher degree of arrhythmia, and slower ICa inactivation in electrophysiological assays and a prolonged relaxation phase in video recordings. Our model provides a comprehensive cellular phenotype for HCM with a mutation in a non-sarcomeric protein.

iPSC-derived cardiomyocytes have proved to be a great tool to study genetic cardiac diseases for disease modelling and drug screening [30,42]. One of the biggest problems with disease modelling has been obtaining the appropriate type of control cells. The problem has been the definition of a normal heart and also the potential role of the whole genome in modifying the effect of the mutation in the models [43]. Recent advances in gene editing technologies, such as CRISPR/Cas9 [44], have enabled precise comparisons of disease phenotypes and responses to drugs between the patient iPSC-derived cells and genetically corrected iPSC-derived cells in lines differing from each other only at the mutation site. Here, we generated isogenic hiPSC-CMs as the control cells by correcting the JPH2 p.(Thr161Lys) mutation using CRISPR/Cas9. Using isogenic hiPSC-CMs, we confirmed that the disease phenotype observed in T161K hiPSC-CMs is related to the JPH2 p.(Thr161Lys) mutation. JPH2 mutations linked to HCM have been observed in multiple studies with animal models and human tissues [8,16,17,18,19,20,21]. To our knowledge, our study is the first to model HCM due to mutations in non-sarcomeric proteins with patient-derived iPSC-CMs. Finally, our data highlights the advantage of using isogenic control hiPSC-CMs in modelling HCM.

A central role in contractile function, cell growth, signal transduction, and cell proliferation is played by the actin cytoskeleton [45]. Impairments in the torsional mechanics underlying heart failure may be due to altered sarcomeres of the ventricular cardiomyocytes and alteration of the actin cytoskeleton has been shown to impair cardiac systolic/diastolic function [46,47]. We conducted immunostaining studies to show expression of key proteins in our hiPSC-CMs. With the T161K hiPSC-CMs, similar to previous HCM studies, we could demonstrate enlarged cell and nucleus sizes and sarcomeric disarray compared with isogenic hiPSC-CMs [35,48,49,50], indicating remodeling of the cytoskeleton and compromised contraction machinery in T161K hiPSC-CMs.

To evaluate the functionality, we performed patch clamp and video microscopy. Similar to this and our previous results [35,51], APDs were found to be longer in HCM-specific hiPSC-CMs in other studies [52,53]. The repolarization time (QT-interval) in ECG corresponding to APD in single cells has been reported to be longer in HCM patients [54,55]. Additionally, the prolonged contraction–relaxation duration from video recording in T161K hiPSC-CMs is in line with APD prolongation. We demonstrated that slower inactivation of the ICa was not only responsible for the longer APD but also manifested the arrhythmias in T161K hiPSC-CMs. Previous studies have also shown that slower inactivation of ICas caused longer a APD [56] and that it was an arrhythmogenic substrate [57].

In this study, one of the limitations is the immaturity of the hiPSC-CMs and how they functionally, metabolically, and structurally more resemble fetal than adult CMs. The reliability of studies with hiPSC-derived CMs can be increased in the future by further developing the differentiation and maturation protocols.

## 5. Conclusions

In summary, our findings demonstrate that T161K hiPSC-CMs can recapitulate the cellular phenotype of HCM that is caused by a mutation in a non-sarcomeric gene (the JPH2 gene with the p.(Thr161Lys) mutation). In our study, we found differences in the morphology and electrophysiology, as well as in the beating properties between the T161K and isogenic hiPSC-CMs. These findings verified the potential of using hiPSC-CMs and CRISPR/Cas9 genome editing for a better understanding of genetic cardiac diseases.

## Figures and Tables

**Figure 1 biomedicines-11-01558-f001:**
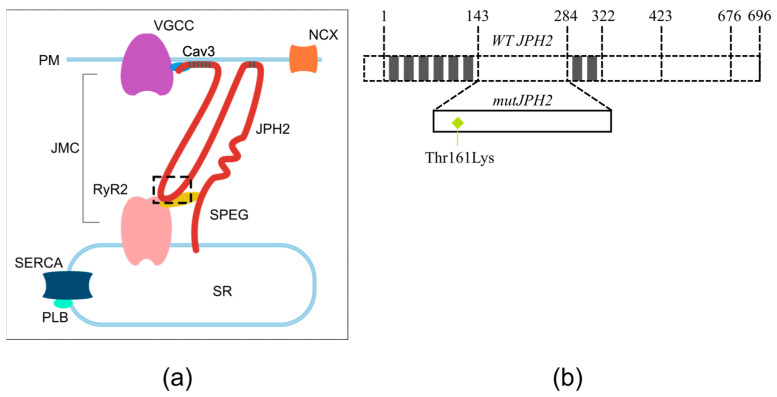
Localization of JPH2 in cardiomyocytes; (**a**) Overview of human JPH2 localized in the junctional membrane complex (JMC) of a cardiomyocyte. The highlighted region is the joining region that contains the mutation JPH2-Thr161Lys; (**b**) Schematic in the right-side panel represents the JPH2 protein topology (1–696 amino acids), which contains the N terminus, an N-terminal membrane occupation, and recognition nexus (MORN) region repeating eight times along the cell membrane phospholipids (1–143 and 284–322). Between the MORN-motifs I and II, the joining region (143–284) resides in direct interaction with the ryanodine-2 receptor (RyR2) [16]. The zoom-in on the joining region shows the JPH2-Thr161Lys mutation. The following structure consists of an alpha helix (322–423), divergent region (423–676), and a transmembrane domain (676–696) in the C-terminal that anchors the JPH2 to the ER/SR to maintain proximity to the PM. SR, Sarcoplasmic Reticulum; PLB, Phospholamban; SERCA, Sarco/Endoplasmic Reticulum Ca^2+^-ATPase; SPEG, Striated Muscle Enriched Protein Kinase; RyR2, Ryanodine Receptor 2; JMC, Junctional Membrane Complex; JPH2, Junctophilin-2; PM, Plasma Membrane; VGCC, Voltage-Gated Calcium Channel; Cav3, Caveolin-3; NCX, Sodium–Calcium Exchanger.

**Figure 2 biomedicines-11-01558-f002:**
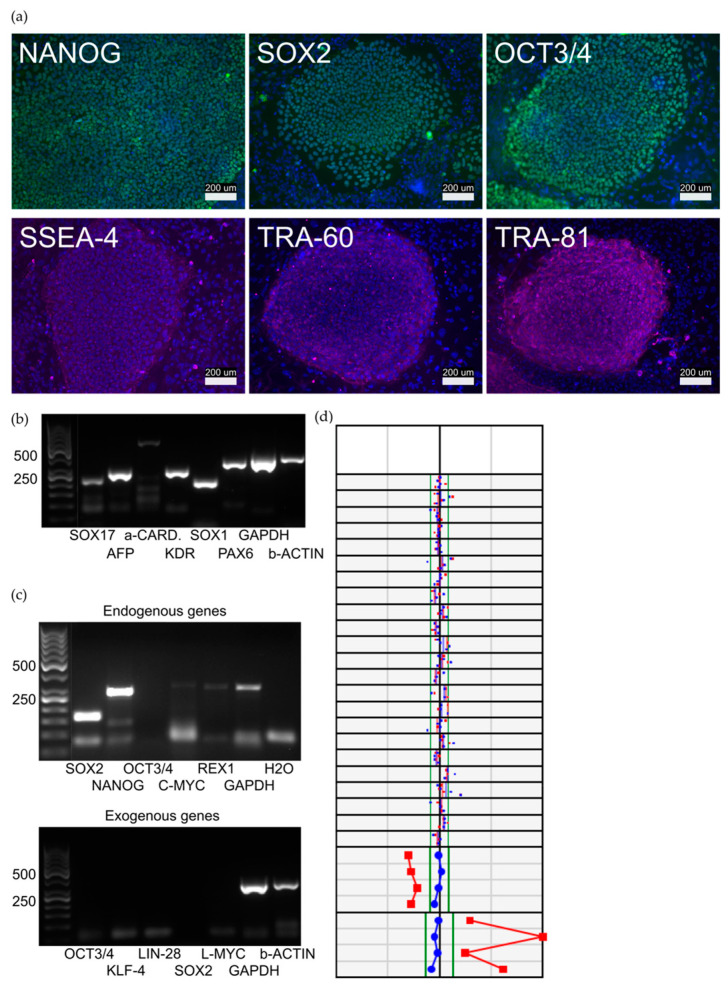
Characterization of the 09703.HCMJp hiPSC line. (**a**) The hiPSCs formed colonies expressing Nanog, Sox2, Oct3/4, SSEA-4, TRA-60, and TRA-81. Scale bars 200 µm. (**b**) hiPSCs expressed endogenous SOX2 (151 bp), Nanog (287 bp), OCT3/4 (144 bp), cMYC (328 bp), and REX1 (306 bp). Virally transferred Sendai exogenes exo-OCT4 (483 bp), exo-KLF4 (410 bp), exo-SOX2 (451 bp), and exo-cMYC (532 bp) were absent in hiPSCs. (**c**) As a proof of pluripotency, hiPSCs were differentiated into EBs, which expressed markers from all germ layers: endoderm (SOX17 (120 bp), AFP (209 bp)), mesoderm (α-cardiac actin (486 bp), KDR (218 bp)), and ectoderm (SOX1 (166 bp), PAX6 (274 bp)). GAPDH (302 bp) was used as a housekeeping control in each PCR experiment. (**d**) The hiPSC line was karyotypically normal (46, XY) in the KaryoLite BoBs Assay (PerkinElmer). Karyotype analyses of the 09703.HCMJp cell line. Red and blue dots indicate chromosomal signal ratios of sample DNA against female (red) and male (blue) reference normal karyotype DNA. Signal from normal chromosomes against both male and female references should lie inside the reference area around value 1, whereas with an abnormal karyotype both signals lie outside the reference area. A female probe pattern is defined when X and Y probe ratios are included in the expected range for a female sample (red line/dots inside and blue line/dots outside the normal expected X/Y range); a male pattern is defined by a reverse pattern (blue line/dots inside and red line/dots outside the normal expected X/Y range). Each plot shows the signal of two technical replicates of the same sample.

**Figure 3 biomedicines-11-01558-f003:**
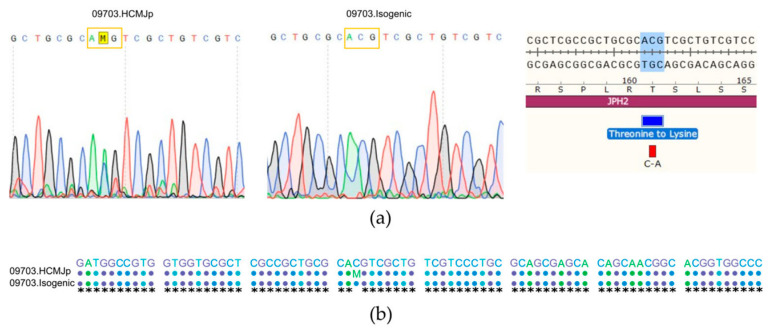
Generation of the isogenic control cell line and localization of JPH2 in cardiomyocytes; (**a**) The heterozygous mutation JPH2 c.482C > A,p. (Thr161Lys) in the mutated CMs was corrected using CRISPR-CAS9 genome editing. Validation of the corrected isogenic line by sanger sequencing is shown. The mutated amino acid codon is boxed and M represents adenosine or cytosine in the patient line; (**b**) The sequence alignment of the reference sequence, patient line 09703.HCMJp, and the isogenic line is shown.

**Figure 4 biomedicines-11-01558-f004:**
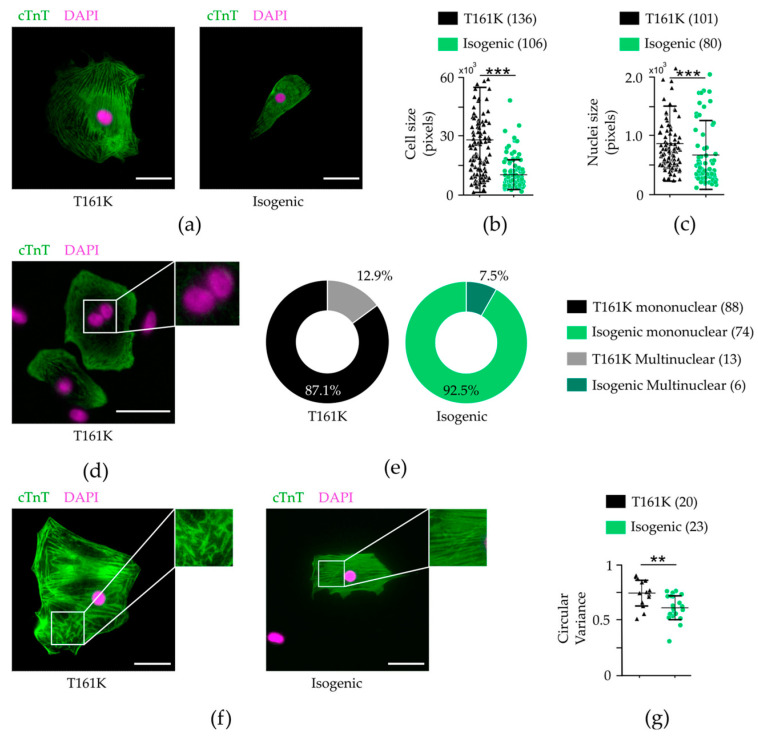
Phenotypic characterization of T161K and isogenic hiPSC-CMs; (**a**) The cell size of hiPSC-derived CMs after 37 days of culture as single cells. Representative immunofluorescent images of JPH2 hiPSC-CMs (left) and isogenic hiPSC-CMs (right) stained with antibodies for cTnT (green) and DAPI (magenta). Scale bars 100 µm. JPH2 hiPSC-CMs present with significantly larger (**b**) cell bodies and (**c**) nuclei sizes than isogenic hiPSC-CMs; (**d**) Representative immunofluorescent image stained with antibodies for cTnT (green) and DAPI (magenta), showing multinucleation (white frame) in hiPSC-CMs. Scale bars 100 µm; (**e**) T161K hiPSC-CMs have a larger fraction of multinucleated (blue) hiPSC-CMs than the other the analyzed cells; (**f**) Representative immunofluorescent images of hiPSC-CMs with misaligned (left) and aligned sarcomeres stained with antibodies for cTnT (green) and DAPI (magenta). Scale bars 100 um; (**g**) T161K hiPSC-CMs have increased sarcomeric disarray. Data represent mean ± SE. ** *p* < 0.01, *** *p* < 0.001; Mann–Whitney U-test. Numbers in parentheses represent the number of cells used.

**Figure 5 biomedicines-11-01558-f005:**
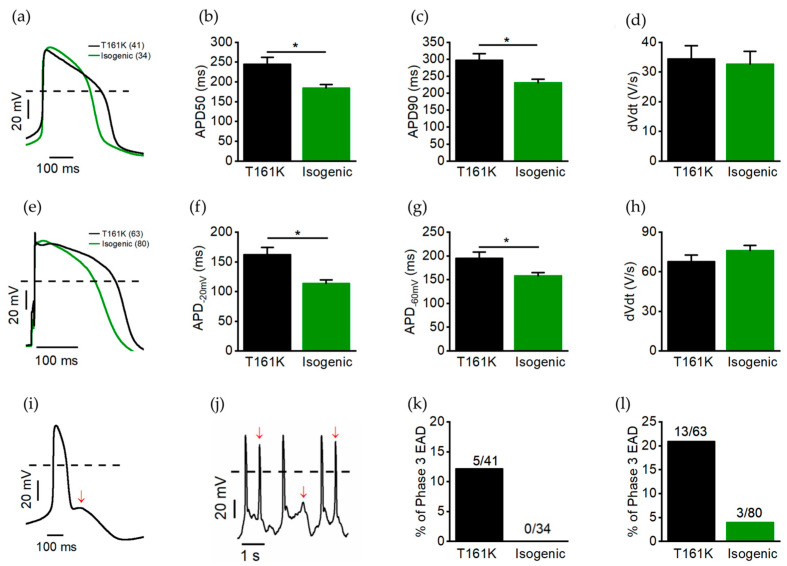
Baseline action potential properties in T161K and isogenic hiPSC-CMs; (**a**) Representative spontaneous AP traces from T161K hiPSC-CMs and isogenic hiPSC-CMs. (**b**,**c**) Averaged values for action potential duration (APD) at 50% (APD50) and 90% (APD90) repolarization. (**d**) Average values for upstroke velocity (dVdt). (**e**) The hiPSC-CMs were stimulated at 1 Hz, representative traces of stimulated APs are shown. (**f**,**g**) Average values for APD at −20 mV (APD −20 mV) and at −60 mV (APD −60 mV). (**h**) Average values for upstroke velocity (Vmax). (**i**) Representative AP trace exhibiting phase 3 EAD and (**j**) phase-3-EAD-induced triggered arrhythmias indicated by red arrows. (**k**) Percentage of occurrence of phase 3 EAD during spontaneous beating. (**l**) Percentage of occurrence of phase 3 EAD during stimulation at 1 Hz. Data are presented as mean ± S.E:M. * *p* < 0.05; unpaired *t*-test. Numbers in parentheses represent the number of cells used.

**Figure 6 biomedicines-11-01558-f006:**
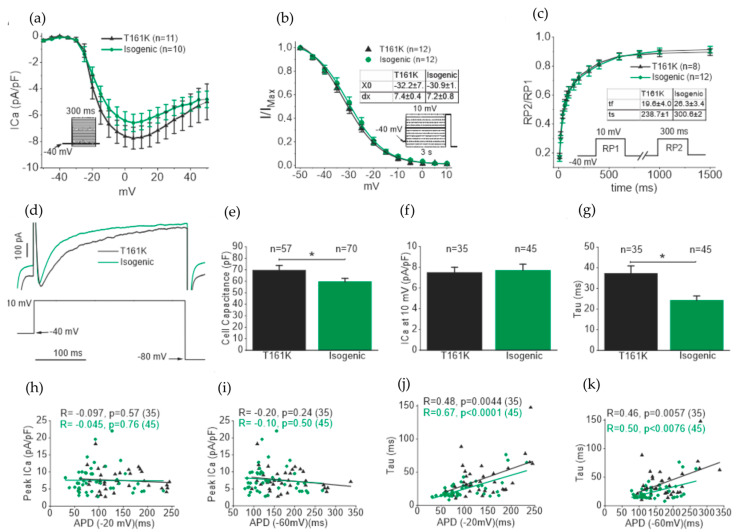
Calcium current (ICa) in T161K and isogenic hiPSC-CMs. (**a**) Current density–voltage relationship of ICas recorded from T161K and isogenic hiPSC-CMs. (**b**) Voltage-dependent inactivation curve of ICas. (**c**) Average time course of recovery from inactivation of ICa. Peak calcium currents elicited by P2 were normalized (I2/I1) and plotted as a function of the recovery interval. (**d**) ICa responses to step voltage protocol. (**e**) Comparison of cell capacitances (* *p* < 0.05, unpaired *t*-test). (**f**) Comparison of ICa current densities at 10 mV. (**g**) Comparison of decay time constant of ICa obtained from monoexponential fitting. (* *p* < 0.05, unpaired *t*-test). (**h**,**i**) Correlation test (Pearson correlation) between ICa current densities and APD (−20 mV) and APD (−60 mV) (Pearson Correlation). (**j**,**k**) Correlation test between time constant and APD (−20 mV) and APD (−60 mV). Numbers in parentheses represent the number of cells used. For fitting procedures, see text. Inset: voltage-clamp protocol used.

**Figure 7 biomedicines-11-01558-f007:**
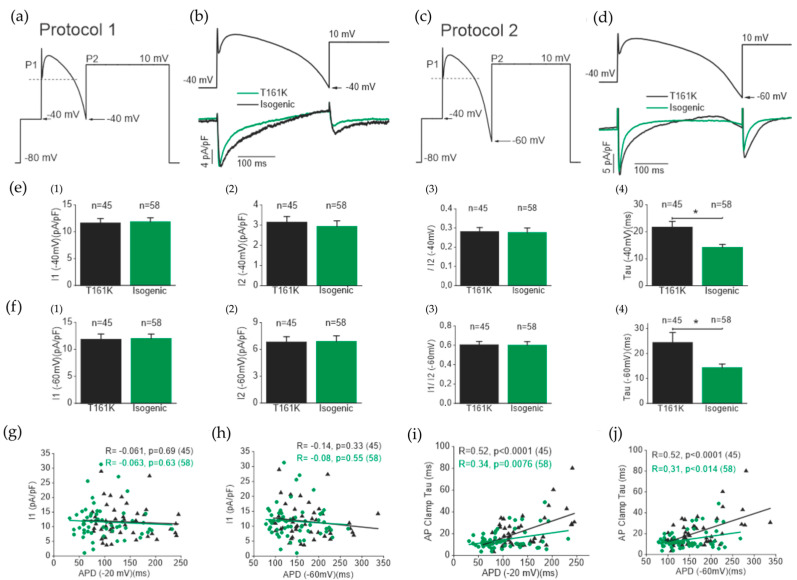
Action potential clamp (APC) study in T161K and isogenic hiPSC-CMs. (**a**) The modified protocol for the APC technique to study the calcium current elicited at 10 mV from −40 mV. (**b**) Under action potential clamping conditions (above panel), the ICa elicited by P1 and P2 originated at −40 mV (below panel). (**c**) The modified protocol for the APC technique to study the calcium current elicited at 10 mV from −60 mV. (**d**) Under action potential clamping conditions (above panel), the ICa elicited by P1 and P2 originated at −60 mV (below panel). (**e**) Comparison of ICa current densities in response to protocol A (e1) P1, (e2) P2, (e3) I1/I2, and (e4) time constant (* *p* < 0.05, unpaired *t*-test). (**f**) Comparison of ICa current densities in response to protocol C (f1) P1, (f2) P2, (f3) I1/I2, and (f4) time constant (* *p* < 0.05, unpaired *t*-test). Correlation test (Pearson correlation) between the ICa current densities obtained from protocol A and (**g**) APD (−20 mV) and (**h**) APD (−60 mV). Correlation test between the time constant of ICa inactivation obtained from protocol A and (**i**) APD (−20 mV) and (**j**) APD (−60 mV).

**Figure 8 biomedicines-11-01558-f008:**
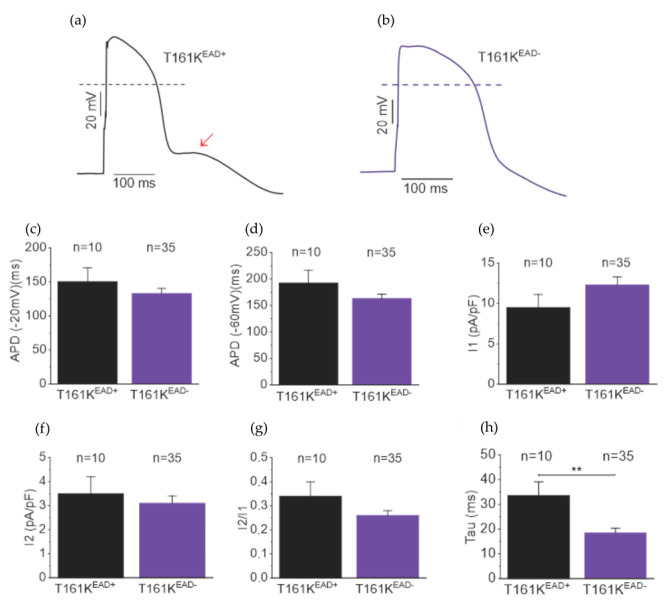
Phase 3 EAD in the T161K hiPSC-CMs. Representative action potential (AP) traces of T161K hiPSC-CMs (**a**) with and (**b**) without the presence of phase 3 EAD indicated by red arrows. Comparison of (**c**) APD (−20 mV) and (**d**) APD (−60 mV) between T161K hiPSC-CMs with and without presence of phase 3 EAD. Comparison of (**e**) I1, (**f**) I2, and (**g**) I2/I1 elicited by using the protocol 1 action potential clamp between T161K hiPSC-CMs with and without presence of phase 3 EAD. (**h**) Comparison of the time constant of calcium currents elicited by using the protocol 1 action potential clamp between T161K hiPSC-CMs with and without the presence of phase 3 EAD. ** *p* < 0.01; paired *t*-test.

**Figure 9 biomedicines-11-01558-f009:**
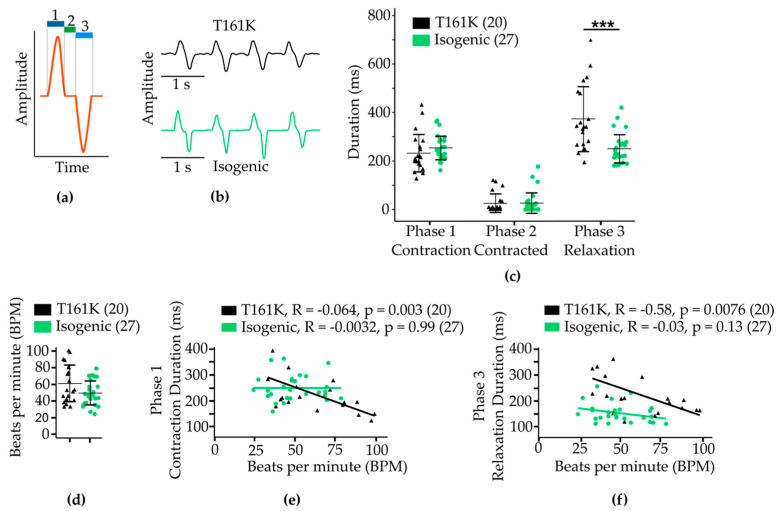
Functional phenotyping of contractility; (**a**) Illustration of a beating signal. 1: Time of contraction (phase 1), 2: Time contracted (phase 2), and 3: Time of relaxation (phase 3); (**b**) Representative beating traces of T161K and isogenic hiPSC-CMs; (**c**) Durations of beating phases 1–3; (**d**) Beating rate; (**e**) Absolute time of contraction against BPM; (**f**) Absolute time of relaxation against BPM. Data are presented as mean ± SE. Numbers in parentheses represent the number of cells used. *** *p* < 0.001.

## Data Availability

The data underlying this article will be shared upon reasonable request to the corresponding author.

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
