# Peer review of "The Junctophilin-2 Mutation p.(Thr161Lys) Is Associated with Hypertrophic Cardiomyopathy Using Patient-Specific iPS Cardiomyocytes and Demonstrates Prolonged Action Potential and Increased Arrhythmogenicity"

_biomedicines, 2023, doi:10.3390/biomedicines11061558_

Round 1

Reviewer 1 Report

This article by Valtonen et al is well written and interesting. The authors have shown that induced pluripotent stem cells derived cardiomyocytes can be used to study genetically transmitted cardiac diseases, such as hypertrophic cardiomiopathy. To this end, they have used skin fibroblasts from a male obstructive hypertrophic cardiomiopathy patient. The paper is interesting from a clinical cardiologist’s point of view, but I recommend that it also be reviewed by a geneticist.

Author Response

Thank you for the comment. In our manuscript, one of the authors is working in a company, Blueprint Genetics, which offers genetic testing and advice. He has been involved in manuscript preparation and reviewing process and the company also carried out the genetic testing in our work.

Reviewer 2 Report

In the manuscript, Valtonen and co-workers describe the establishment of a novel cellular model to study hypertrophic cardiomyopathy. The paper is interesting, but the approach used is not outbreaking. In my opinion, the introduction should be used to further discussed existing previously used approaches, both for cardiomyopathies or other diseases. For example, the paper 10.1186/s13287-022-02905-0 recently summarized iPSC approaches base methodologies for hypertrophic cardiomyopathies, and it should be cited and discussed. Also, many interesting models have been recently developed for other diseases, and the paper would be more precious if these were commented on and the benefits associated with them or with the cell model described in the paper were analyzed. To this purpose, please see: 10.3390/ijms23095105 10.1002/cpt.1832, 10.1016/j.mcp.2018.01.001

As for the results, my main concern is related to the CRISPR-Cas9 procedure. It seems that there is no control for the genomic editing, since the mutated cell line did not undergo any control procedure. Thus the authors compare two cell lines (the isogenic and the mutant) that experienced very different conditions, even if they derive from the same cells, in principle. My question is how did they reduce the possible influence by the CRISPR/Cas9 procedure on the results. The authors should also specify better the selection conditions (was G418 kept in the culture medium after the 24 hours at a lower concentration?).

Author Response

Point 1: In my opinion, the introduction should be used to further discussed existing previously used approaches, both for cardiomyopathies or other diseases. For example, the paper 10.1186/s13287-022-02905-0 recently summarized iPSC approaches base methodologies for hypertrophic cardiomyopathies, and it should be cited and discussed.

Response 1: Thank you for the comment and this has now been corrected as suggested to include relevant citations and discussion (lines 75-78).

Point 2: Also, many interesting models have been recently developed for other diseases, and the paper would be more precious if these were commented on and the benefits associated with them or with the cell model described in the paper were analyzed. To this purpose, please see: 10.3390/ijms23095105 , 10.1002/cpt.1832, 10.1016/j.mcp.2018.01.001

Response 2: Thank you for the comment. This has now been corrected as suggested to include relevant citations and discussion. (lines 64-74)

Point 3: As for the results, my main concern is related to the CRISPR-Cas9 procedure. It seems that there is no control for the genomic editing, since the mutated cell line did not undergo any control procedure. Thus the authors compare two cell lines (the isogenic and the mutant) that experienced very different conditions, even if they derive from the same cells, in principle. My question is how did they reduce the possible influence by the CRISPR/Cas9 procedure on the results. The authors should also specify better the selection conditions (was G418 kept in the culture medium after the 24 hours at a lower concentration?).

Response 3: Thank you for the comment. Traditionally we have used iPSCs from a healthy individual as control cells, but in this current paper we corrected the mutation with gene editing to be used as the control. However, in our currently ongoing experiments we have also included a wild type control cell line and the CRISPR-Cas9 corrected cell line behaves the same way as the wild type control. Regarding the selection conditions, we have now expanded the paragraph for better clarity (lines 155-156).

Reviewer 3 Report

Current report applied an isogenic counterpart generated by the CRISPR/Cas9 genome editing method to explore the abnormal functions in JPH2-HCM, skin fibroblasts from a Finnish patient with JPH2 p.(Thr161Lys). This is an interesting study and it needs to conduct the concerns below.

1.      You derived iPSCs from a patient heterozygous that must follow the ethics of medical practice.

2.      The hiPSC cell line 09703.HCMJp the manufacturer belonged to unknown.

3.      In Figure 5, sample size in each group was not indicated.

4.      Cases with phenotype of HCM caused by mutation in a non-sarcomeric gene were not introduced in clear.

5.      Merit(s) of current finding seems ignored. Why?

6.      Limitation(s) of current report were also not conducted.

Quality of language is good. But, it will be better to check through a professional editing.

Author Response

Point 1: You derived iPSCs from a patient heterozygous that must follow the ethics of medical practice.

Response 1: This is important and the ethics committee statement for the study has been presented at the statement section of the manuscript as instructed by the journal guidelines (lines 592-596).

Point 2: The hiPSC cell line 09703.HCMJp the manufacturer belonged to unknown.

Response 2: Description for the cell line source information has been stated at the Materials and Methods section and the iPSC lines have been generated in our laboratory. The hiPSC lines have been registered to hPSCreg -database (https://hpscreg.eu/) with IDs TAUi001-A and TAUi001-A-1.

Point 3: In Figure 5, sample size in each group was not indicated.

Response 3: This has now been amended to include sample sizes in all required sections of Figure 5.

Point 4: Cases with phenotype of HCM caused by mutation in a non-sarcomeric gene were not introduced in clear.

Response 4: This has now been added in Introduction and presented more clearly (line 41-44).

Point 5: Merit(s) of current finding seems ignored. Why?

Response 5: Thank you for the important comment. The importance of the findings has been already previously mentioned throughout the paper, but they have now been emphasized better and in more detail (lines 87-91, 531-533, 546-548, 554-557, 563-565, 573-575, 577-578).

Point 6: Limitation(s) of current report were also not conducted.

Response 6: Thank you for the comment. Limitations have now been properly presented in Discussion of the manuscript as instructed by the journal (lines 567-570).  

Round 2

Reviewer 2 Report

acceptable for publication in the present form